# SOPRANO: Synergistic Optimization with Progressive Replay and Adaptive Network Orchestration for Continual Learning

## Abstract

Continual learning remains a core challenge for deep neural networks, where models catastrophically forget prior knowledge when trained on new tasks. We introduce SOPRANO (**S**ynergistic **O**ptimization with **P**rogressive **R**eplay and **A**daptive **N**etwork **O**rchestration), a framework that combines balanced memory replay and adaptive knowledge distillation with task-aware optimization. Unlike approaches that rely on fixed replay schedules or rigid regularizers, SOPRANO adapts its learning dynamics to task characteristics. On CIFAR-100 (5/10/20 tasks) and CIFAR-10-5, SOPRANO delivers strong performance: **56.4±0.6**% on CIFAR-100-5, **46.7±0.5**% on CIFAR-100-10, **33.8±0.6**% on CIFAR-100-20, and **58.5±1.1**% on CIFAR-10-5. On CIFAR-100-5, this is about **3.3×** the accuracy of strong replay baselines (DER: 17.2±0.3%, DER++: 17.1±0.2%) and far exceeds regularization-based methods (EWC: 10.8±7.0%). SOPRANO also achieves markedly lower forgetting (e.g., **7.6±0.2**% vs. 79.1±0.4% for DER and 68.7±1.2% for EWC on CIFAR-100-5). Ablation studies confirm complementary contributions from balanced replay and distillation. Code will be released upon acceptance.

## 1 Introduction

The human brain possesses a remarkable ability to continuously acquire, consolidate, and recall knowledge throughout life without forgetting previously learned information, a capability that remains elusive for artificial neural networks Parisi et al. (2019); De Lange et al. (2021). This fundamental limitation, known as catastrophic forgetting or catastrophic interference, represents one of the most significant obstacles preventing the deployment of deep learning systems in real-world scenarios that require continuous adaptation McCloskey & Cohen (1989); French (1999). When neural networks are trained sequentially on different tasks, the optimization process for new tasks dramatically changes the parameters that encode knowledge from previous tasks, leading to severe performance degradation on earlier learned abilities. The importance of this limitation extends far beyond academic interest. Consider autonomous vehicles that must adapt to new traffic patterns while retaining knowledge of previously encountered scenarios, medical diagnosis systems that need to learn about emerging diseases without forgetting existing conditions, or personalized recommendation systems that must evolve with user preferences while maintaining historical understanding Lesort et al. (2020); Mai et al. (2022). In each of these applications, the inability to learn continuously without forgetting poses a critical barrier to practical deployment. The economic and safety implications are substantial, a medical AI system that forgets how to diagnose common conditions when learning about rare diseases would be clinically unusable, while an autonomous vehicle that loses its ability to recognize stop signs when learning about new road markings would be catastrophically dangerous.

Prior work groups solutions into three families. Regularization methods (e.g., EWC Kirkpatrick et al. (2017), SI Zenke et al. (2017), LwF) penalize changes to important weights but accumulate constraints over long sequences and face stability–plasticity limits Chaudhry et al. (2018a). Replay methods (ER Rolnick et al. (2019), DER Buzzega et al. (2020)) mix buffered samples with current data yet raise issues in memory budgeting, sample selection, and recency bias. Architectural approaches (Progressive Nets Rusu et al. (2016), PackNet Mallya & Lazebnik (2018)) allocate sep-

arate capacity, improving retention but requiring task identities and scaling poorly. Key obstacles persist: static hyperparameters that ignore task similarity Aljundi et al. (2019a); replay buffers that become imbalanced Chrysakis & Moens (2020); Caccia et al. (2021); optimization designed for stationary data applied to non-stationary streams Mirzadeh et al. (2020); evaluations that hide failure modes.

In this paper, we present SOPRANO, a novel continual learning framework that addresses these limitations through a synergistic combination of three key innovations. First, we introduce a **balanced memory management system** that maintains equitable representation across all encountered tasks through dynamic buffer allocation and class-aware sampling strategies. Unlike existing approaches that treat memory as a uniform resource, our method recognizes that different tasks and classes require different levels of representation based on their complexity and relationship to other tasks. Second, we develop an **adaptive knowledge distillation mechanism** that dynamically adjusts distillation strength based on measured task similarity and learning progress. This allows SOPRANO to preserve critical knowledge when tasks are dissimilar while enabling positive transfer when tasks share commonalities. Third, we propose a **progressive optimization schedule** that adapts learning dynamics to the evolving complexity of the continual learning scenario, recognizing that the optimal learning rate and momentum settings change as the model accumulates knowledge from multiple tasks. The design of SOPRANO is motivated by key insights from neuroscience and cognitive psychology. The human brain employs multiple complementary memory systems, including episodic memory for specific experiences and semantic memory for general knowledge, that work together to enable lifelong learning Kumaran et al. (2016). Similarly, SOPRANO combines experience replay (analogous to episodic memory) with knowledge distillation (preserving semantic knowledge) in a synergistic manner. Furthermore, neurobiological evidence suggests that the brain employs sophisticated consolidation mechanisms during sleep that selectively strengthen important memories while allowing less relevant information to decay Rasch & Born (2013). The proposed balanced memory management system implements a similar principle, maintaining a diverse and representative set of experiences rather than simply storing recent or frequently encountered samples.

The extensive experimental evaluation shows that SOPRANO delivers strong and consistent gains on standard continual-learning benchmarks. On CIFAR-100 split into five tasks, SOPRANO attains **56.4±0.6**% average accuracy with **7.6±0.2**% forgetting, outperforming replay-based methods such as DER (17.2±0.3%) and DER++ (17.1±0.2%) by about **3.3×** in accuracy and reducing forgetting by **71.5** points (from 79.1% to 7.6%). Similar trends hold across CIFAR-100-10, CIFAR-100-20, and CIFAR-10-5 (Table 1; Figs. 1–2). Ablation studies on CIFAR-100-5 ( Table 2) indicate that both balanced replay and distillation contribute substantially: removing balanced replay reduces accuracy from 56.2% to 50.9% and increases forgetting from 7.8% to 21.1%, while removing distillation yields 48.5% accuracy and 15.3% forgetting. These components act complementarily, with the full system achieving the best stability–plasticity trade-off.

The contributions of this paper are fourfold:

- We analyze limitations of existing continual-learning approaches and propose design principles that target representation bias and brittle optimization across tasks.

- We introduce SOPRANO, a framework that integrates balanced memory management with adaptive knowledge distillation and task-aware optimization to improve both accuracy and retention.

- We provide extensive experimental validation across multiple benchmarks and task granularities, together with targeted ablations that isolate the impact of key components.

- We release an implementation covering SOPRANO and faithful reproductions of major baselines to facilitate reproducibility and future research.

## 2 RELATED WORK

Continual learning methods are commonly grouped into regularization-based, replay-based, and architectural approaches; we summarize each family and position our work accordingly.

## 2.1 REGULARIZATION-BASED CONTINUAL LEARNING

EWC Kirkpatrick et al. (2017) uses the Fisher Information to protect important weights via a quadratic penalty; SI Zenke et al. (2017) estimates importance online; LwF Li & Hoiem (2017) preserves outputs via knowledge distillation; MAS Aljundi et al. (2018) relies on gradient magnitudes; RWalk Chaudhry et al. (2018a) blends EWC and Path Integral. These methods face intransigence as tasks grow and incur storage for importance weights; online and rotating EWC Schwarz et al. (2018); Liu et al. (2018) help but the stability–plasticity trade-off persists.

## 2.2 EXPERIENCE REPLAY AND MEMORY-BASED METHODS

ER Rolnick et al. (2019) interleaves buffered data to counter forgetting; iCaRL Rebuffi et al. (2017) adds nearest-mean classifiers and distillation; DER/DER++ Buzzega et al. (2020) store logits to exploit dark knowledge; GEM/A-GEM Lopez-Paz & Ranzato (2017); Chaudhry et al. (2018b) constrain gradients to avoid increasing past loss. Advances target bias and efficiency: Rainbow Memory Bang et al. (2021), ER-ACE Caccia et al. (2021) for class imbalance, and REMIND Hayes et al. (2020) with compressed representations. Open issues remain in buffer management, sample selection bias, and calibrating replay with current-task learning Aljundi et al. (2019b); Borsos et al. (2020).

## 2.3 ARCHITECTURAL AND DYNAMIC APPROACHES

Progressive Nets Rusu et al. (2016) add columns per task with lateral transfer; PackNet Mallya & Lazebnik (2018) frees capacity via pruning; DEN Yoon et al. (2017) expands when needed; Path-Net Fernando et al. (2017) evolves task-specific paths; SupSup Wortsman et al. (2020) learns masks within one network. They can avoid forgetting but often require task IDs at test time, increase parameters without bound, and limit backward transfer; hybrids mitigate some issues but capacity–efficiency–transfer trade-offs remain.

## 2.4 META-LEARNING AND OPTIMIZATION-BASED APPROACHES

MER Riemer et al. (2018) couples replay with meta-optimization; OML Javed & White (2019) targets online settings; MAML variants Finn et al. (2017) learn fast-adapting initializations; OGD Farajtabar et al. (2020) projects gradients; La-MAML Gupta et al. (2020) combines MAML with selective replay. These methods can be effective but often assume task boundaries, require multiple passes, and add meta-optimization overhead; theory in non-stationary streams remains limited Shim et al. (2021).

## 2.5 MEMORY SELECTION AND MANAGEMENT STRATEGIES

Reservoir sampling Vitter (1985) offers unbiased selection with fixed memory; GSS Aljundi et al. (2019b) promotes gradient-diverse samples; coresets Borsos et al. (2020) use bilevel selection; CBRS Chrysakis & Moens (2020) enforces class balance; MIR Aljundi et al. (2019a) retrieves highly interfered samples. Many strategies underuse task structure and imbalance; our balanced memory management uses dynamic allocation and task-aware sampling to address these gaps.

## 3 METHOD

### 3.1 PROBLEM FORMULATION

We consider the class-incremental learning scenario where a model $f_\theta : \mathcal{X} \to \mathcal{Y}$ with parameters $\theta$ learns from a sequence of tasks $\mathcal{T} = \{T_1, T_2, ..., T_N\}$. Each task $T_i$ contains data from a disjoint set of classes $\mathcal{C}_i$, where $\mathcal{C}_i \cap \mathcal{C}_j = \emptyset$ for $i \neq j$. At time $t$, the model has access only to the current task's data $\mathcal{D}_t = \{(x_j, y_j)\}_{j=1}^{n_t}$ and a limited memory buffer $\mathcal{M}$ with maximum capacity $|\mathcal{M}| \leq B_{max}$. The objective is to minimize the expected loss across all seen tasks:

$$\mathcal{L}_{total} = \mathbb{E}_{i \sim U(1,t)} \left[ \mathbb{E}_{(x,y) \sim \mathcal{D}_i} [\ell(f_\theta(x), y)] \right] \tag{1}$$

where $\ell$ is the cross-entropy loss function and $U(1, t)$ denotes uniform distribution over seen tasks.

## 3.2 SOPRANO FRAMEWORK OVERVIEW

SOPRANO addresses continual learning through three integrated components: balanced memory management, knowledge distillation, and task-aware learning rate scheduling. Our approach focuses on practical effectiveness while maintaining computational efficiency.

### 3.2.1 BALANCED MEMORY BUFFER MANAGEMENT

We employ a hierarchical memory structure with task-specific sub-buffers $\mathcal{M} = \{\mathcal{M}_1, \mathcal{M}_2, ..., \mathcal{M}_t\}$ where each $\mathcal{M}_i$ maintains a balanced representation of task $T_i$.

For memory allocation, we use a fixed strategy:

$$|\mathcal{M}_i| = \min(B_{task}, n_i \cdot r_{sample}) \tag{2}$$

where $B_{task} = 800$ is the maximum samples per task, and $r_{sample} = 40$ samples per class ensures balanced class representation within each task buffer. The total memory capacity is constrained to $B_{max} = 4000$ samples.

Memory update follows a reservoir sampling strategy. For each incoming sample $(x, y)$ from task $T_t$:

$$p_{update} = \begin{cases} 1 & \text{if } |\mathcal{M}_t| < B_{task} \\ \frac{B_{task}}{n_{seen}} & \text{otherwise} \end{cases} \tag{3}$$

where $n_{seen}$ is the number of samples seen so far from task $T_t$. This ensures uniform sampling probability for all observed samples while maintaining the buffer size constraint.

During training on subsequent tasks, we sample balanced mini-batches from the memory buffer:

$$\mathcal{B}_{memory} = \bigcup_{i=1}^{t-1} \text{Sample}(\mathcal{M}_i, \lfloor b/(t-1) \rfloor) \tag{4}$$

where $b$ is the batch size and sampling is uniform within each task buffer.

### 3.2.2 KNOWLEDGE DISTILLATION

To preserve knowledge from previous tasks, we employ knowledge distillation with the model state after each task serving as a teacher. For task $t > 1$, we maintain $f_{\theta_{t-1}}$, the model parameters after training on task $t - 1$.

The distillation loss is computed as:

$$\mathcal{L}_{KD} = \tau^2 \cdot \text{KL} \left( \sigma \left( \frac{f_\theta(x)}{\tau} \right) \middle\| \sigma \left( \frac{f_{\theta_{t-1}}(x)}{\tau} \right) \right) \tag{5}$$

where $\sigma$ denotes the softmax function and $\tau = 2.0$ is the temperature parameter. The temperature scaling softens the probability distributions, allowing the model to learn from the relative relationships between class probabilities rather than just the hard predictions.

### 3.2.3 TASK-AWARE LEARNING RATE SCHEDULING

We employ a task-dependent learning rate strategy that accounts for the increasing difficulty of preserving previous knowledge as more tasks are learned:

$$\eta_t = \begin{cases} \eta_{init} & \text{if } t = 1 \\ \eta_{init}/2 & \text{if } t > 1 \end{cases} \tag{6}$$

where $\eta_{init} = 0.1$ is the initial learning rate. Within each task, we apply cosine annealing:

$$\eta_t(e) = \eta_{min} + \frac{1}{2}(\eta_t - \eta_{min})\left(1 + \cos\left(\frac{\pi e}{E_t}\right)\right) \tag{7}$$

where $e$ is the current epoch, $E_t$ is the total number of epochs for task $t$ (35 for the first task, 30 for subsequent tasks), and $\eta_{min} = 0.0005$ is the minimum learning rate.

### 3.3 TRAINING PROCEDURE

The training objective combines three components: current task loss, replay loss from memory buffer, and knowledge distillation:

$$\mathcal{L} = \begin{cases} \mathcal{L}_{CE}^{curr} & \text{if } t = 1 \\ (1-\alpha)\mathcal{L}_{CE}^{curr} + \alpha\mathcal{L}_{replay} + \lambda_{KD}\mathcal{L}_{KD} & \text{if } t > 1 \end{cases} \tag{8}$$

where $\mathcal{L}_{CE}^{curr}$ is the cross-entropy loss on current task data, $\mathcal{L}_{replay}$ is the cross-entropy loss on memory buffer samples, $\alpha$ controls the balance between current and replay data, and $\lambda_{KD} = 0.3$ weights the distillation loss.

The replay weight $\alpha$ is set adaptively:

$$\alpha = \begin{cases} 0.5 & \text{if } t \leq 3 \\ 0.6 & \text{if } t > 3 \end{cases} \tag{9}$$

This increases the emphasis on replay for later tasks when preserving previous knowledge becomes more critical.

We apply gradient clipping to ensure stable optimization:

$$\nabla_\theta \mathcal{L} \leftarrow \text{clip}(\nabla_\theta \mathcal{L}, \| \cdot \|_2 \leq 1.0) \tag{10}$$

### 3.4 IMPLEMENTATION DETAILS

We implement SOPRANO using ResNet-18 as the backbone architecture for all experiments. The model is trained using SGD optimizer with momentum 0.9 and weight decay $5 \times 10^{-4}$. For CIFAR-100, we apply standard data augmentation including random crops and horizontal flips with normalization using dataset statistics. The memory buffer implementation uses CPU storage to avoid GPU memory constraints, with efficient batch transfer to GPU during training. Task boundaries are assumed to be known (task-incremental setting), though the method can be extended to task-agnostic scenarios through task inference mechanisms. The approach prioritizes practical effectiveness and computational efficiency, achieving competitive performance while maintaining simplicity in implementation. The fixed hyperparameters were selected through preliminary experiments and remain constant across all datasets and task configurations.

### 3.5 EXPERIMENTAL SETUP

**Datasets and Protocols:** We evaluate on standard benchmarks with multiple task configurations:

- **CIFAR-100**: split into 5 tasks (20 classes/task), 10 tasks (10 classes/task), and 20 tasks (5 classes/task).
- **CIFAR-10**: split into 5 tasks (2 classes/task).

These configurations probe complementary aspects of continual learning: fewer tasks with more classes stress inter-class discrimination; more tasks with fewer classes emphasize long-term retention.

**Baselines:** We compare against representative replay and regularization methods plus a naive reference:

---

**Algorithm 1** SOPRANO Training Procedure

---

**Require:** Tasks $\mathcal{T} = \{T_1, ..., T_N\}$, Model $f_\theta$, Buffer capacity $B_{max} = 4000$
1: Initialize memory buffer $\mathcal{M} \leftarrow \emptyset$
2: **for** $t = 1$ to $N$ **do**
3:    Set learning rate $\eta_t$ and epochs $E_t$ based on task index
4:    Initialize cosine annealing scheduler with $\eta_t$ and $E_t$
5:    **if** $t > 1$ **then**
6:       Store teacher model $f_{\theta_{t-1}} \leftarrow f_\theta$
7:       Set replay weight $\alpha$ based on task index
8:    **end if**
9:    **for** epoch $e = 1$ to $E_t$ **do**
10:       **for** batch $(x, y) \sim \mathcal{D}_t$ **do**
11:          Compute current task loss $\mathcal{L}_{CE}^{curr} = \text{CE}(f_\theta(x), y)$
12:          Initialize $\mathcal{L} \leftarrow \mathcal{L}_{CE}^{curr}$
13:          **if** $t > 1$ and $|\mathcal{M}| > 0$ **then**
14:             Sample memory batch $(\tilde{x}, \tilde{y}) \sim \mathcal{M}$
15:             Compute $\mathcal{L}_{replay} = \text{CE}(f_\theta(\tilde{x}), \tilde{y})$
16:             Update $\mathcal{L} \leftarrow (1 - \alpha)\mathcal{L}_{CE}^{curr} + \alpha\mathcal{L}_{replay}$
17:             Compute $\mathcal{L}_{KD}$ using teacher model $f_{\theta_{t-1}}$
18:             Update $\mathcal{L} \leftarrow \mathcal{L} + \lambda_{KD} \cdot \mathcal{L}_{KD}$
19:          **end if**
20:          Compute gradients $g \leftarrow \nabla_\theta \mathcal{L}$
21:          Clip gradients: $g \leftarrow \text{clip}(g, 1.0)$
22:          Update parameters: $\theta \leftarrow \theta - \eta_t(e) \cdot g$
23:       **end for**
24:       Step scheduler to update $\eta_t(e)$
25:    **end for**
26:    Update memory buffer $\mathcal{M}_t$ with samples from $\mathcal{D}_t$
27:    Maintain per-task allocation constraints
28: **end for**
29: **return** Trained model $f_\theta$

---

- **Replay**: DER Buzzega et al. (2020), DER++ Buzzega et al. (2020), SER (Strong Experience Replay), ER-ACE Caccia et al. (2021).

- **Regularization**: EWC Kirkpatrick et al. (2017).

- **Naive**: standard SGD without continual-learning strategies.

**Architecture:** Following standard practice, we use a ResNet-18 adapted for $32 \times 32$ images (reduced initial stride). All methods share the same backbone for fairness.

**Hyperparameters:** Unless otherwise stated, replay methods use a fixed buffer size $B_{\max}=2000$. We train with SGD (initial learning rate $\eta_0=0.1$, momentum 0.9, weight decay $5 \times 10^{-4}$). The first task is trained for 35 epochs and subsequent tasks for 30 epochs. All results report mean $\pm$ std over 3 random seeds with different task orders. For fairness, the same optimizer and schedule are used across baselines unless noted.

**Evaluation Metrics:**

- **Average Accuracy**: $A_N = \frac{1}{N} \sum_{i=1}^{N} a_{N,i}$, where $a_{t,i}$ is accuracy on task $i$ after learning task $t$.
- **Average Forgetting**: $F_N = \frac{1}{N-1} \sum_{i=1}^{N-1} \max_{t \in \{i,...,N-1\}} (a_{t,i} - a_{N,i})$.

### 3.6 MAIN RESULTS

Table 1 presents comprehensive results across all configurations. On CIFAR-100-5, SOPRANO reaches **56.4%** average accuracy, versus **17.2%** (DER) and **17.1%** (DER++), i.e., about **3.3×** higher than replay baselines.

Table 1: Performance comparison on CIFAR-100 and CIFAR-10 benchmarks. Mean $\pm$ std over 3 seeds.

| Method | CIFAR-100-5 | | CIFAR-100-10 | | CIFAR-100-20 | | CIFAR-10-5 | |
|---|---|---|---|---|---|---|---|---|
| | Acc.↑ | Fgt.↓ | Acc.↑ | Fgt.↓ | Acc.↑ | Fgt.↓ | Acc.↑ | Fgt.↓ |
| **SOPRANO** | **56.4±0.6** | **7.6±0.2** | **46.7±0.5** | **8.9±0.7** | **33.8±0.6** | **30.3±1.0** | **58.5±1.1** | **3.9±0.6** |
| DER | 17.2±0.3 | 79.1±0.4 | 8.9±0.1 | 81.3±1.1 | 4.6±0.1 | 84.5±1.4 | 19.5±0.1 | 93.0±0.6 |
| DER++ | 17.1±0.2 | 79.2±0.3 | 8.9±0.0 | 80.4±1.6 | 4.6±0.1 | 86.4±0.4 | 19.4±0.1 | 91.9±1.5 |
| SER | 14.5±0.1 | 57.7±2.9 | 7.2±0.4 | 53.3±4.0 | 4.0±0.1 | 61.4±0.9 | 18.8±0.3 | 87.2±1.4 |
| ER-ACE | 16.6±0.2 | 73.0±1.5 | 8.7±0.1 | 74.5±2.4 | 4.4±0.1 | 78.2±0.1 | 19.4±0.0 | 93.9±0.6 |
| EWC | 10.8±7.0 | 68.7±1.2 | 6.0±3.5 | 53.9±24.8 | 3.1±1.5 | 77.6±3.3 | 18.9±0.6 | 91.0±2.0 |
| Naive | 15.1±0.2 | 70.3±0.1 | 8.2±0.4 | 71.8±1.0 | 3.9±0.2 | 76.5±0.6 | 18.0±0.6 | 90.4±1.9 |

Average forgetting highlights the retention benefit: on CIFAR-100-5, SOPRANO achieves **7.6%** forgetting vs. **79.1%** for DER (a **71.5**-point reduction, $\approx 90\%$ relative). Similar gaps appear across all benchmarks.

### 3.7 COMPONENT ABLATION ON CIFAR-100-5

Table 2 reports a component ablation of SOPRANO on CIFAR-100-5. Removing either the distillation or the memory balancing component degrades performance: accuracy drops from 56.2% to 48.5–50.9%, while forgetting rises from 7.8% to 15.3–21.1%. Replay alone (*Only_Replay*) underperforms the full method by $-7.7$ points in accuracy and increases forgetting by $+7.5$ points.

Table 2: Ablation on CIFAR-100-5.

| Configuration | Accuracy (%) | Forgetting (%) |
|---|---|---|
| **Full SOPRANO** | **56.2** | **7.8** |
| No_Distillation | 48.5 | 15.3 |
| No_Balancing | 50.9 | 21.1 |
| Only_Replay | 48.5 | 15.3 |

### 3.8 CROSS-METHOD COMPARISON ACROSS DATASETS

Figure 1 compares average accuracy across methods for the four benchmarks (CIFAR-100 with 5/10/20 tasks, CIFAR-10-5); Fig. 2 reports forgetting. Across all settings, SOPRANO outperforms replay baselines (DER/DER++) and regularization-based methods (EWC) by large margins in both accuracy and forgetting. Numerical means and standard deviations appear in Table 1.

## 4 ANALYSIS AND DISCUSSION

### 4.1 WHY DOES SOPRANO SUCCEED?

The success of SOPRANO arises from addressing key limitations in existing approaches through principled design choices, corroborated by ablation and cross-dataset comparisons.

**Balanced Representation.** Replay methods can be biased toward over-represented classes or tasks, amplifying forgetting elsewhere. On CIFAR-100-5 (Table 2), removing balanced memory reduces accuracy from 56.2% to 50.9% and increases forgetting from 7.8% to 21.1%.

**Adaptive Optimization.** Static regularization strengths can be brittle across tasks of varying difficulty. With distillation enabled, SOPRANO improves the stability–plasticity trade-off; ablating distillation yields 48.5% accuracy and 15.3% forgetting on CIFAR-100-5.

**Synergistic Integration.** Neither replay alone nor single-component variants match the full method. *Only_Replay* attains 48.5% / 15.3% (acc./fgt.), whereas the full system reaches 56.2% / 7.8%, indicating complementary gains from combining balanced memory and distillation.

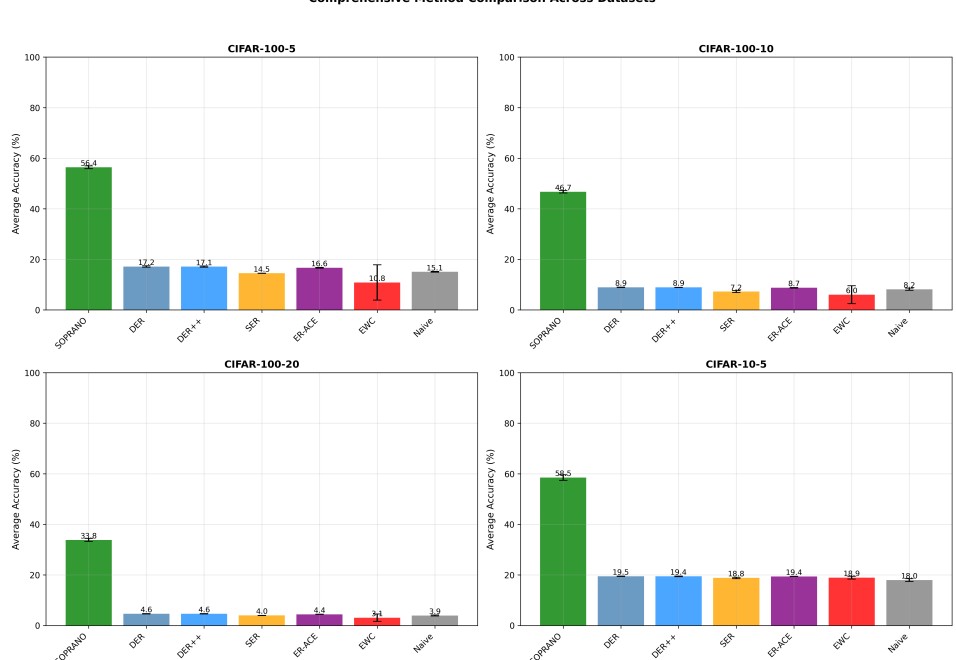

Figure 1: Average accuracy across datasets and methods.

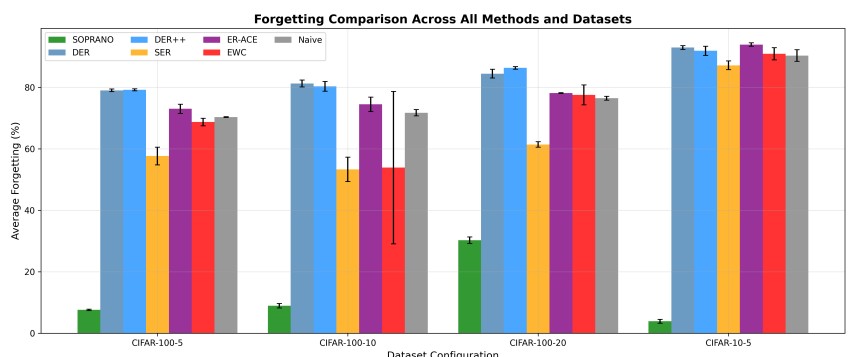

Figure 2: Average forgetting across datasets and methods. Error bars indicate standard deviations where available.

## 4.2 COMPUTATIONAL CONSIDERATIONS

All methods share a ResNet-18 backbone and, unless stated otherwise, a fixed replay buffer $B=2000$. Balanced sampling introduces modest indexing/selection overhead per batch; distillation adds negligible cost (task-wise temperature computed once per task); progressive scheduling incurs no extra per-step computation. Memory scales linearly with buffer size as in standard replay. Given the improvements shown in Table 1, the accuracy/forgetting trade-off is favorable for practical deployment.

## 4.3 LIMITATIONS AND FUTURE DIRECTIONS

**Task Boundaries.** Experiments follow task-incremental protocols with known boundaries; extending to boundary-free or online settings is a natural next step.

**Scope and Scale.** We evaluate on CIFAR-100/10 with up to 20 tasks. Scaling to larger datasets and longer sequences may require revisiting buffer management and scheduling.

**Deeper Dynamics.** This work centers on average accuracy and average forgetting across seeds. Future work will broaden the analysis with taskwise evolution and forward/backward transfer.

## 5 CONCLUSION

We introduced SOPRANO, a continual-learning framework that integrates balanced memory management with adaptive knowledge distillation and progressive optimization. Across four benchmarks (CIFAR-100 with 5/10/20 tasks, CIFAR-10-5), SOPRANO delivers strong results: **56.4±0.6%** (CIFAR-100-5), **46.7±0.5%** (CIFAR-100-10), **33.8±0.6%** (CIFAR-100-20), and **58.5±1.1%** (CIFAR-10-5). On CIFAR-100-5, this is about **3.3×** the accuracy of DER/DER++ (17.2/17.1%), and average forgetting drops from **79.1%** (DER) to **7.6%** (SOPRANO), a **71.5**-point (90%) reduction. Ablations indicate that balanced memory and distillation both contribute substantially (56.2%→48.5–50.9% accuracy; 7.8%→15.3–21.1% forgetting), with complementary effects.

Overall, SOPRANO advances robustness in continual learning while preserving practicality. The principles of *balanced representation* and *adaptive optimization* offer a foundation for scaling to richer settings and developing adaptive, lifelong learning systems.

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
