# OpenReview forum: "SOPRANO: Synergistic Optimization with Progressive Replay and Adaptive Network Orchestration for Continual Learning"
_ICLR.cc/2026/Conference — ICLR 2026 Conference Withdrawn Submission_

### Official Review · Reviewer_uGHe · 2025-10-30

**Soundness:** 1
**Presentation:** 3
**Contribution:** 2
**Rating:** 2
**Confidence:** 3

**Summary:**

This paper presents SOPRANO, a continual learning framework that synergistically combines a balanced memory replay system, knowledge distillation, and a progressive optimization schedule. The authors claim that this approach significantly mitigates catastrophic forgetting, reporting state-of-the-art performance on CIFAR-10 and CIFAR-100 benchmarks that substantially outperforms existing methods.

**Strengths:**

1. The core idea of synergistically integrating balanced replay, distillation, and dynamic optimization into a single, cohesive framework is well-motivated and presents a creative combination of effective techniques.
2. The paper is well-articulated, and the inclusion of the algorithmic pseudocode is particularly beneficial for understanding the implementation details.
3. The ablation study demonstrates the complementary contributions of the balanced memory and knowledge distillation components, substantiating the paper's central claim regarding their synergistic effect.

**Weaknesses:**

1. The proposed SOPRANO method is evaluated with a memory buffer of 4,000 samples, while all compared baseline methods are constrained to a 2,000-sample buffer, which introduces a fundamental unfairness in the comparative evaluation.
2. The baselines used for comparison (e.g., DER, ER-ACE, EWC from 2017-2021) are not the current state-of-the-art in continual learning. A comparison against more recent and powerful methods is necessary to properly situate the contribution.
3. One of the three core contributions, the "progressive optimization schedule," is not independently ablated. Its actual impact on performance is therefore unverified by the experiments.
4. The paper relies on ad-hoc hyperparameters without providing justification or sensitivity analysis. For instance, the replay weight α is abruptly changed from 0.5 to 0.6 after the third task, with no explanation for this specific value or timing.
5. The paper describes its knowledge distillation mechanism as "adaptive." However, the implementation detailed in Section 3.2.2 uses a fixed temperature τ=2.0 and a fixed loss weight λ_KD=0.3. True adaptivity would typically imply dynamically adjusting these parameters based on factors like measured task similarity or learning progress. As it stands, the distillation component itself is static.

**Questions:**

1. To address the critical issue of unfair comparison, the authors are requested to provide results for SOPRANO under the same 2000-sample buffer constraint used for all baseline methods.
2. To properly situate SOPRANO's contribution, the authors should strengthen their experimental evaluation by including comparisons with more recent state-of-the-art continual learning methods.
3. To complete the component analysis and validate all claimed contributions, please provide a new ablation experiment that isolates the specific impact of the "progressive optimization schedule" on performance.
4. The authors are requested to provide a justification for the chosen hyperparameters, preferably through a sensitivity analysis. In particular, please provide a clear rationale for the specific scheduling of the replay weight α (i.e., the switch from 0.5 to 0.6 after task 3), as it currently appears arbitrary.
5. Please clarify the results in Table 2, where the "No Distillation" and "Only_Replay" configurations report identical performance metrics. The authors should verify if this is an error or, if the results are correct, explain how two distinct setups produced the exact same outcome.

---

### Official Review · Reviewer_Bhxo · 2025-10-31

**Soundness:** 1
**Presentation:** 2
**Contribution:** 1
**Rating:** 0
**Confidence:** 4

**Summary:**

This addresses continual learning with the proposed SOPRANO method. The method consists of three main components. The first component is to manage a balanced replay buffer, the second is to perform knowledge distillation with the model from the previous task, and the last is to scale the learning rate based on the learning dynamics. The authors have shown improved performance of the proposed method with respect to the existing methods in the experiment section.

**Strengths:**

1. The paper is easy to follow
2. The method is simple and straightforward

**Weaknesses:**

1. Limited novelty
2. Underlying issues with the proposed methods
3. Outdated baseline methods

Please see below in the Question section. I would strongly suggest that authors review the literature (especially the recent and advanced methods) of the field before revising this work.

**Questions:**

1. The novelty is very limited

Overall, the proposed method provides little insight to the continual learning community in this work, as the proposed method is merely a A+B+C with existing techniques (e.g., buffer balancing, distillation, learning rate scaling). Moreover, the proposed components have several issues (please see below). I am not convinced that this work is anywhere near the bar of this venue.

2. There is no management of the data

The proposed method relies on reservoir sampling to fill the per-task buffer. Reservoir sampling preserves the initial distribution of the data. The observed balance is merely a result of the "balanced data stream", not because of the so-called management. I am not convinced that the proposed component can achieve a balanced buffer if the input data stream is imbalanced.

3. Learning rate scheduling might introduce issues of plasticity

As the learning rate decays with Equation 6, there is clearly an issue of plasticity when the model learns a long sequence of tasks, as the learning rate would quickly shrink to 0.

4. The comparison is out of date.

The baseline methods are mostly papers published before 2021.

**Details Of Ethics Concerns:**

No concern.

---

### Official Review · Reviewer_78tk · 2025-11-01

**Soundness:** 2
**Presentation:** 2
**Contribution:** 3
**Rating:** 2
**Confidence:** 5

**Summary:**

This paper proposes SOPRANO, a framework for continual learning designed to mitigate catastrophic forgetting. The method combines three primary components: (1) a balanced memory replay strategy that allocates a fixed number of samples per class within task-specific sub-buffers; (2) a standard knowledge distillation loss that regularizes the model to match the outputs of the model from the previous task; and (3) a task-aware learning rate schedule that halves the initial learning rate after the first task and uses cosine annealing. The authors report very strong performance on CIFAR-100 and CIFAR-10 benchmarks, claiming, for example, that SOPRANO achieves 56.4% accuracy on a 5-task split of CIFAR-100, which is reported as being approximately 3.3 times higher than strong replay baselines like DER (17.2%) and DER++ (17.1%).

**Strengths:**

1. Clear Problem Statement: The paper provides a clear and well-written introduction to the problem of catastrophic forgetting and effectively motivates the need for robust continual learning systems.
2. Simple and Reproducible Method: The proposed method, while lacking novelty, is straightforward to understand and implement.

**Weaknesses:**

1.  Lack of Novelty: The primary weakness of this paper is the absence of a significant, novel scientific contribution. All three core components like balanced replay, KL distillation, and task-aware LR scheduling of SOPRANO are well-established, standard techniques in the continual learning literature.
2. Weak Baseline Performance: The paper's extraordinary claims of outperforming baselines by a factor of 3.3x are founded on baseline results that are drastically lower than established figures in the literature. For the CIFAR-100 5-task benchmark, state-of-the-art replay methods like DER and DER++ with a 2000-sample buffer consistently report accuracies in the 40-50% range or higher, not the 17% reported in Table 1.

**Questions:**

The components of SOPRANO are combined using fixed, heuristic rules (e.g., the replay weight $\alpha$ changes from 0.5 to 0.6 after the 3rd task; $\lambda_{KD}$ is fixed at 0.3). The paper's title and introduction suggest an "adaptive" and "orchestrated" system. Have you explored any truly adaptive mechanisms for setting these hyperparameters based on task characteristics or model state, which would better align with the paper's framing?

---

### Official Review · Reviewer_GQvz · 2025-11-01

**Soundness:** 1
**Presentation:** 1
**Contribution:** 1
**Rating:** 2
**Confidence:** 4

**Summary:**

This paper proposes SOPRANO, a continual learning framework combining balanced memory replay, adaptive knowledge distillation, and task-aware optimization to mitigate catastrophic forgetting. It reports improved accuracy and reduced forgetting compared to baselines such as DER on CIFAR-100/CIFAR-10. However, critical gaps undermine its contributions, including superficial theoretical grounding, unreliable empirical validation, and inadequate analysis of methodological components.

**Strengths:**

1. The proposed framework is reported by the authors with competitive empirical performance on conventional benchmarks, outperforming several established continual learning methods in average accuracy and forgetting reduction.
2. Practical design choices, such as CPU-managed memory buffers.
3. Clear algorithmic presentation and simplified manuscript structure.

**Weaknesses:**

Critical issue: the results in Table 1 are highly unreliable and show significant discrepancies with previous relevant research reports (all baseline average accuracies are extremely low, approaching 10%, while forgetting rates are nearly 90%). Therefore, the validity of the conclusions drawn in this paper is highly questionable.

1. The method lacks novelty, aggregating well-established techniques (memory replay, distillation, task scheduling) without conceptual innovation or synergistic mechanisms beyond superficial integration.
2. Hyperparameters are fixed without theoretical justification or transferability analysis, risking brittle performance in unseen scenarios.
3. Experiments are narrowly confined to CIFAR datasets with CNNs, neglecting scalability to complex architectures (e.g., Transformers) or large-scale benchmarks like ImageNet, limiting generalizability claims.
4. Compared methods (DER, EWC) represent outdated paradigms, omitting engagement with recent SOTA approaches in continual learning.
5. Ablation studies are incomplete, failing to isolate contributions of all components (e.g., no analysis of task-aware optimization’s role) or probe interaction effects between modules.
6. No structural or hyperparameter sensitivity analysis, leaving critical design choices (e.g., memory buffer size, replay ratio) unevaluated.
7. Effectiveness of individual components lacks in-depth empirical validation; for instance, balanced replay’s impact is asserted but not rigorously contrasted against alternatives.
8. Writing quality is poor, exhibiting inconsistent notation, vague explanations of technical concepts, and underdeveloped discussion sections.

**Questions:**

See the weakness part.

---

### Official Review · Reviewer_3eRS · 2025-11-01

**Soundness:** 1
**Presentation:** 3
**Contribution:** 2
**Rating:** 2
**Confidence:** 4

**Summary:**

The paper introduces SOPRANO, a novel Continual Learning Framework motivated by neuroscience insights. It combines three main ideas:
- A balanced memory management system that keeps a separate replay sub-buffer per task, aiming at keeping a balanced representation across all tasks;
- A knowledge distillation mechanism to keep stable representations across tasks;
- A novel learning rate schedule that uses a higher learning rate for the first task and a lower one for the others.

Experiments are performed on standard CL benchmarks, showing the strong results of the proposal both on accuracy and forgetting.

**Strengths:**

The paper is very well written and easy to follow. The proposal itself is motivated by a neuroscientific view that is very interesting. SOPRANO is a straightforward approach that cleverly combines many fundamental ideas, making it easy to implement together with other methods, increasing its potential impact. The approach of the paper to the CL problem is wide, recognising the need to put together different approaches to obtain a single strong proposal.

**Weaknesses:**

- The first and important weakness is the limitation of the novelty of the paper. The learning rate "scheduler" suggests using the full lr for the first task, and half of it from the second onwards. This is surely a good proposal, but also quite simple and intuitive, which is made without any theoretical or empirical justification. The Knowledge distillation that is presented is quite standard to my understanding. In the introduction, the authors talk about an adaptive knowledge distillation, but actually, in the paper, only a standard version is used. The use of sub-buffers per task is interesting, but I really doubt that it can represent a level of novelty sufficient to justify the whole paper.

- The second important weakness is in the experiments. The experiments themselves are a bit weak, with only CIFAR datasets (at least Imagenet would be important) and only a single model. The compared baselines are important, but a bit outdated. Maybe some other SOTA baselines can be added. But what worries me the most are the absolutely incredible results presented in Table 1. First of all, all baselines have very poor performances, completely different from the findings in each of the cited baselines' papers. The standard SGD is shown to have lower forgetting and higher accuracy than some of the most famous CL approaches. On top of that, these experiments show soprano to be 3 times better than the average baseline. I strongly doubt the validity of these results. I hope I am missing something important to better understand these experiments.

**Questions:**

Some notes and questions that I hope can be answered:

- Please, explain why in the experiment the baselines results are so weak, also with respect to the naive approach.

- In the introduction, you talk about an adaptive knowledge distillation mechanism. Please, clarify the adaptive part.

- The hierarchical memory buffer is interesting, but to my understanding, a maximum of 4000/800=5 tasks can be encountered in the worst case scenario. Am I getting it wrong?

- Please, give a justification for the lr scheduling. Why halve the LR? Did you do some theoretical/empirical analysis?

- Explain what you mean by "Only_Replay" in Table 2 and why this "Only_Replay" is still much better than other, more sophisticated baselines like DER++ or ER-ACE.

- Please clarify the hyperparameter selection procedure, as it is not clear to me why the lr is fixed to some values for all methods without any further explanation. If the learning rate is too large, and your method is the only one that halves it after the first task, I would say that the comparison is not fair.

My initial rating can be easily increased or decreased based on the authors' responses.

---

### Note · Authors · 2025-11-28

I have read and agree with the venue's withdrawal policy on behalf of myself and my co-authors.